# Current and Future Perspectives on the COVID-19 Vaccine: A Scientometric Review

**DOI:** 10.3390/jcm11030750

**Published:** 2022-01-29

**Authors:** Alireza Noruzi, Behzad Gholampour, Sajad Gholampour, Somayeh Jafari, Razieh Farshid, Agata Stanek, Ali Akbar Saboury

**Affiliations:** 1Department of Knowledge and Information Science, Faculty of Management, University of Tehran, Tehran 1417935840, Iran; noruzi@ut.ac.ir (A.N.); behzad903727@yahoo.com (B.G.); jafari.somayyeh@gmail.com (S.J.); 2Department of Sport Management, Faculty of Physical Education and Sports Sciences, Kharazmi University, Tehran 3197937551, Iran; sajad908919@yahoo.com; 3Department of Knowledge and Information Science, Faculty of Psychology and Education, Kharazmi University, Tehran 3197937551, Iran; razieh.farshid@gmail.com; 4Department of Internal Medicine, Angiology and Physical Medicine, Faculty of Medical Sciences in Zabrze, Medical University of Silesia, Batorego 15 St., 41-902 Bytom, Poland; 5Institute of Biochemistry and Biophysics, University of Tehran, Tehran 1417614335, Iran

**Keywords:** COVID-19 vaccine, COVID-19 vaccination, bibliometric analysis, research trends, research topics, scientific collaboration

## Abstract

This study attempted to draw the present and future perspective of the COVID-19 vaccine by identifying the most important scientists and their scientific contexts, trends of research topics, and relationships between different entities. Methods: To achieve this purpose, bibliometric and scientometric techniques were used to analyze 6288 scientific documents contributing to COVID-19 vaccines from the beginning of 2019 to 13 December 2021, indexed in the Web of Science. Results: The United States (US) had the greatest impact by publishing 2104 documents and receiving 32,958 citations. The US and the UK countries had the highest level of scientific collaborations with 192 collaborative studies. The University of Oxford and the Harvard Medical School were the most active institutions, and the University of Oxford and Emory University were the most influential institutions. Pollard AJ and Lambe T had the most publications and the highest citations and h-index. T Lambe, SC Gilbert, M Voysey, and AJ Pollard from the University of Oxford had the highest number of co-authorships. More than 19% of the research was conducted in the field of immunology. *The Vaccines* journal had the most publications, with 425 articles. The US Department of Health & Human Services granted the most research. In 2019, studies were focused on the topics of COVID-19 virus identification and ways to deal with it; in 2020, studies focused on the topics of COVID-19 and vaccines, whereas in 2021, they focused on the topics of COVID-19 vaccines and their effects, vaccines hesitancy, the role of healthcare workers in COVID-19, as well as discussions about these vaccines in the social media. Conclusions: Recognition of the most important actors (countries, institutes, researchers, and channels for the release of COVID-19 vaccine studies), research trends, and fields of study on the COVID-19 vaccine can be useful for researchers, countries, and policy makers in the field of science and health to make decisions and better understand these vaccines.

## 1. Introduction

The new coronavirus, or COVID-19, is a deadly virus common between humans and animals [1], which was identified by the World Health Organization on 11 March 2020, as a pandemic [2]. The pandemic showed that it is not only a medical problem. Since the pandemic affects society as a whole, new research frameworks are needed, including interdisciplinary collaborations, to manage the problem [3,4]. It was in December 2019 that the first news of the Huanan seafood market was received in Wuhan City, the capital of Hubei Province, China, which reported cases of the disease [5,6,7,8], a disease that has been the news trend of all news media so far and has occupied the minds of all individuals and researchers in the 21st century. In other words, as of 15 December 2021, according to reports sent to the World Health Organization, 270,791,973 confirmed cases of COVID-19 and 5,318,216 deaths were registered [9]. Therefore, with the rapid spread of COVID-19 and the proliferation of patients around the world, as well as the change in new mutations and the lack of definitive treatment for it, the issue of vaccine and individual vaccination has become a priority of governments as the only way to control the disease [10,11,12]. Therefore, scientists believe that safety and control of this disease are achieved when 60 to 70% of the world’s population is vaccinated against this disease [13].

However, a vaccine is a biological preparation that provides acquired active immunity to a specific disease [11]. The main types of vaccines include live-attenuated vaccine, inactivated vaccine, subunit, recombinant, polysaccharide and conjugate, and toxoid vaccine [14]. To develop a vaccine, complete information on the characteristics of the antigen, adjuvant, vaccine production, and delivery system must be available [15]; due to the availability of genomic and structural information on the new coronavirus [16,17], different vaccines for this virus were produced [14]. In general, for the first time on 11 and 18 December 2020, the United States Food and Drug Administration (FDA) gave the Pfizer/BioNTech and Moderna COVID-19 vaccines an emergency authorization to control the disease [12]. Therefore, based on the latest global statistics, there are 232 vaccine candidates to date. One hundred and thirteen vaccines are in clinical trials on humans, 75 vaccines are in the preclinical phase, and approximately 15 vaccines have received the necessary licenses and are used by countries [15]. Therefore, according to reports received from the World Health Organization about vaccination, until 13 December 2021, a total of 8,200,642,671 doses of vaccine have been injected [9]. In total, in addition to vaccines and vaccinations, regular hand washing, mask use, observance of social distance, and adherence to health protocols have been declared the most reliable, efficient, and effective preventive measures against the new coronavirus.

In the section on research and technologies in the field of vaccines, we are witnessing the emergence of interdisciplinary fields, research collaborations, and new research fronts to transform the future of science and technology in this field. Identifying and evaluating research, research areas, and research fronts in COVID-19 vaccines familiarize researchers and related organizations with current and future research trends in this field and help them to better conduct training programs in this field. The present study helps to identify important interdisciplinary topics and research areas resulting from scientific publications, the most prolific and influential researchers, institutions, countries, and journals that publish new research on coronavirus vaccines and develops a plan for future research collaborations.

Therefore, considering the importance of using vaccines in the prevention and control of pandemics and following an increase of scientific publications on the COVID-19 vaccine, attention to scientometric studies to assess the research productivity and collaborations of various disciplines, researchers, and institutions of the world in this field have gained a special importance. Therefore, the current research aims to draw the present and future perspective of the COVID-19 vaccine studies by identifying the most important actors and their scientific fields, trends in research topics, and relationships between different entities.

Some studies by [2,14,18,19] in particular, examined the field of COVID-19 vaccines from the scientometric and bibliographic perspective. Hence, their studies are consistent with the current research in terms of method and subject. Among the distinguishing points of this study with the research conducted, we can mention the approach of the present study, in which an attempt was made to limit the search process to the beginning of the epidemic, i.e., the period from 2019 to 13 December 2021. Additionally, the intelligent selection of search keywords is another distinctive point of this research, making it possible to analyze studies focused on the COVID-19 vaccines, not the COVID-19 or coronavirus. On the other hand, the reference of the present study to the current and future research fields of studies on the COVID-19 vaccines is another distinguishing point of this research from other studies.

## 2. Materials and Methods

According to the objectives of the present research, the method of bibliometric and scientometric analysis was used to analyze scientific publications on the COVID-19 vaccines. The population of this research consists of all documents on the COVID-19 vaccines from 2019 to 13 December 2021 indexed in the Web of Science database. The methodology of this research was divided into two parts, which are described in detail below.

### 2.1. Data Retrieval

The search strategy was such that in the first step, the Clarivate Analytics Web of Science (WoS) Core Collection was selected [20,21]. Next, the three citation indexes of this database, i.e., SCI-EXPANDED, SSCI, and A&HCI indexes, became the basis of the research. Different names of COVID-19, along with the two words vaccine and vaccination, were searched to collect the documents through the Advanced Search of the WoS, as well as the scientific and common names of different types of vaccines (See Appendix A) in the field of TOPIC (TS phrase in Web of Science) without any restrictions on the TYPE and LANGUAGE of the documents. The search strategy of this study led to the retrieval of 6288 records (See Figure 1), all records were extracted in plain text format in batches of 500 in 13 TXT files and saved on a personal computer; then, all the files were merged into one file for ease in analyzing.

In other words, the website of COVID-19 vaccines www.track.org was used (19 January 2022) to gather information about the type of vaccine, primary developers, and status of approving countries. Additionally, information on the doses purchased from each vaccine was collected from the app.powerbi.com website. The website www.RAPS.org (accessed on 19 January 2022) was also used to identify vaccine-producing countries.

### 2.2. Data Analysis

All extracted files were analyzed using HistCite (v. 12.03.17, Philadelphia, PA, USA), Publish or Perish (v. 8, London, UK), VOSviewer (v. 1.6.17, Leiden, The Netherlands), CiteSpace (v. 5.8.R3, Philadelphia, PA, USA), and Bibliometrix R Package software (v. 3.1.4, Naples, Italy). The analysis process was such that the capabilities of HistCite software were used to analyze information about authors, institutions, countries, and journals [20]. In addition, Bibliometrix R Package was used to draw the maps of geographical distribution, three-field plot, and the process of publishing journals. While VOSviewer was used to determine the status of scientific collaborations between researchers, organizations, and institutions conducting research on the COVID-19 vaccines. Finally, CiteSpace was used to review the research process on the COVID-19 vaccines, as well as the timeline map of countries and institutions in this field.

## 3. Results

Figure 2 shows the trend of published papers on the COVID-19 vaccines during 2019–2021. The results indicate that the trend of publications has increased significantly compared to the beginning years of the COVID-19 virus epidemic, so that the number of related publications has increased from 17 documents in 2019 to 5708 documents in 2021. The widespread outbreak of the COVID-19 virus worldwide, increasing deaths, and more deadly new mutations, have prompted researchers to conduct more research on the production and effectiveness of vaccines to control this disease.

Figure 3 and Table 1 show the co-authorship network and details of the most active authors in COVID-19 vaccine studies. More than 32,800 authors co-authored publications on vaccines. Table 1 lists the authors who have published at least 18 documents in this field. In Figure 3, the nodes represent the authors and links indicate the number of co-authorships formed between the authors. Furthermore, the researchers who collaborated on at least three studies are shown in Figure 3. This network consists of 1491 nodes and 5834 links. T Lambe, SC Gilbert, M Voysey, and AJ Pollard from the University of Oxford had the highest number of co-authorship. According to Table 1, Pollard AJ and Lambe T published most of the publications in this field and are located in the first and second places. However, Pollard AJ and Lambe T, with 3583 and 3579 citations, which are in the first and second places in terms of scientific publications in this field, were able to obtain the highest citation rate and H-index score among their peers. The important point in Table 1 of the year of publication of the authors’ documents is that Wiwanitkit V, with the publication of 23 documents in 2021, is the only active author who has been able to publish this number of documents in this period.

Figure 4 illustrates a three-field plot in the research on the COVID-19 vaccines, focusing on the relationships between authors, countries, and keywords. In Figure 4, rectangles represent each entity (authors on the left, countries in the middle, and keywords on the right). The height of the rectangles refers to the number of studies conducted by countries, authors, and research fields, and the greater the number of studies by each of these entities, the size and height of the rectangles will be more. Lines of communication also reflect the links made between authors and countries, as the more research that was conducted by authors and countries on the subject, the thickness of the links will be greater.

The analysis showed that the US, Italy, China, and the United Kingdom (UK) conducted the most research on COVID-19 vaccines, and the height of the countries’ rectangles indicates that. Furthermore, most of the US studies have focused on COVID-19, COVID-19 Vaccination, COVID-19 Vaccines, Vaccine hesitancy, and COVID-19 Vaccines BNT162B2; these studies have been led by many researchers. Additionally, most Italia research has focused on topics, such as COVID-19, COVID-19 Vaccination, COVID-19 Vaccines, Vaccine hesitancy, COVID-19 Vaccines BNT162B2, and COVID-19 Pandemic; these studies have been conducted mostly by researchers such as AJ Pollard, T Lambe, L Kilmek, PK Aley, and M Voysey. However, Chinese and UK researchers, similarly to American researchers, focused on COVID-19, COVID-19 Vaccination, COVID-19 Vaccines, Vaccine hesitancy, and COVID-19 Vaccines BNT162B2.

Figure 5 and Table 2 show the most active organizations in publishing studies on the COVID-19 vaccines. In total, more than 8000 institutions and universities around the world collaborated independently or in groups. Figure 5 indicates the institutions that cooperated at least in three documents together. This network consists of 1396 nodes and 13,428 links. The results presented in Table 2 and Figure 5 show that many of these studies are the result of collaboration between researchers from European, Asian, and American universities. As shown in Figure 5, the University of Oxford in the UK and the Harvard Medical School in the US, as the most active organizations, played a greater role than other institutions in publishing these studies, and this is visible in Figure 5 and Table 2.

Additionally, based on the results of Table 2, it can be argued that the US and the UK universities played a prominent role in publishing studies on the COVID-19 vaccines. Overall, from the 12 universities listed in Table 2, the US with 6, the UK with 4, and Canada and Israel with one university, were the most participating institutions in this field. However, Table 2 shows that the documents published by the University of Oxford and Emory University have had the greatest citation impact on the scientific community. It should be noted that the 151 studies published by researchers at the University of Oxford received 5546 citations, and the 76 studies published by researchers at the London School of Hygiene & Tropical Medicine obtained 4008 citations.

To analyze the bibliographic information from documents on the COVID-19 vaccines, a three-field plot of institutions, keywords, and journals was designed, which allows us to identify institutions that have studied topics on the COVID-19 vaccines and provides the possibility to identify the field of journals that published such studies. As shown in Figure 6, institutions are on the left, keywords in the middle, and journals on the right. Figure 6 indicates the higher outputs of institutions, such as Oxford University, Harvard Medical Sciences, Tel Aviv University, National and Kapodistrian University of Athens, Johns Hopkins University, Stanford University, and Washington University, which are examined related fields to COVID-19, COVID-19 vaccination, COVID-19 vaccines, public health, COVID-19 vaccine acceptance, and COVID-19 vaccines bnt162b2. Generally, most of the research on vaccines at these institutions has been published in journals of *Vaccines*, *Vaccine*, *Human Vaccines* & *Immunotherapeutics*.

Figure 7 shows the geographical distribution of the research on the COVID-19 vaccines. The results of Figure 7 indicate how the topic of study on the COVID-19 vaccines has been developed on five continents. Perhaps less of a topic addressed by all researchers worldwide, COVID-19 and the development of the vaccine have led to a global collaboration to eradicate it, and researchers in all countries are trying to achieve this by studying and investigating in this field. In Asia, China and India conducted the most research in this field. The US and Canada from the Americas and the UK and Italy from Europe published the highest number of articles on this topic. Australia from Oceania and South Africa from Africa also had the most studies on COVID-19 vaccines.

In total, 122 countries contributed to the publication of studies on the COVID-19 vaccine, and the findings of the most active countries are presented in Table 3. Only four countries in the US, the UK, Italy, and China, with a publication of 61% of the total research conducted, were the most important and main countries producing publications on the COVID-19 vaccines, which played a prominent role in this field. The United States also had the greatest impact on the scientific community in this field by publishing 2104 studies and receiving 32,958 citations. Generally, the efforts of major pharmaceutical companies in the US, the UK, and China to study and investigate the production of the COVID-19 vaccine have been effective in increasing their publications in this field.

Figure 8 shows the geographical distribution of countries participating in the research on the COVID-19 vaccine around the world. As can be seen in Figure 8, the orange links indicate the wide range of collaborations that have taken place over the last three years since the beginning of the COVID-19 epidemic; authors from 153 countries have conducted studies related to the COVID-19 vaccine. Generally, most countries collaborated in publishing these studies so that the US–UK (192), the US–China (116), the US–Canada (95), and the US–Germany (86) have recorded the highest amount of scientific collaborations.

The results of Table 4 show that the *Vaccines* journal with 425 articles and the *Vaccine* journal with 185 articles had the highest number of publications in the field of COVID-19 vaccines. Studies published in the *New England Journal of Medicine* and the *Lancet* had the greatest citation impact on the scientific community, receiving 12,861 citations and 6776 citations, respectively, which could obtain significant citations in this field. The results of Table 4 also indicate that all 14 journals publishing research on the COVID-19 vaccines have an impact factor and are among the qualified journals in their field. According to the results of Table 4, the highest impact factor belongs to the *New England Journal of Medicine* with IF 91,245, and the lowest impact factor belongs to the *International Journal of Environmental Research and Public Health* with IF 3.39. Among them, 10 journals were Q1, 2 journals Q2 and Q3. Furthermore, Table 4 reveals that Nature Publishing Group and MDPI played the most role in this field with two journals.

Continuing to review the results of the publication of studies by journals, Figure 9 depicts the time trend of publication at the beginning of the COVID-19 epidemic in 2019 until now and informs us how to publish studies on the COVID-19 vaccines in journals. Figure 9 shows that since 2020, the trend for publishing studies in the *Vaccines* journal has been on an upward trend, which shows the special attention of this journal to the publication of studies on COVID vaccines. Overall, the journal has increased its number of studies from 13 in 2020 to 411 in 2021. It is followed by *Vaccine* and *Human Vaccines & Immunotherapeutics*, which were other research channels for researchers to publish studies on the COVID-19 vaccines in these years. These journals published 30 and 29 studies in 2020 and 155 and 145 studies in 2021, respectively.

According to the results presented in Table 5, the US Department of Health and Human Services with 437 studies and the National Institutes of Health NIH US with 408 studies had the highest support for the publication of scientific research on the COVID-19 vaccines. Note that the National Institutes of Health (NIH) is an affiliate of the US Department of Health and Human Services that plays a very important role in improving the national health of this country. Therefore, the presence of the National Institute of Allergy Infectious Diseases (NIAID) among the institutions supporting research on the COVID-19 vaccines is self-confirming. However, the European Commission and UK Research Innovation UKRI are in the next level with 169 and 143 studies, respectively. Given the extensive activities of these institutions and organizations, their outstanding efforts and support for research related to this field should not be overlooked because this support had a great impact on the production and manufacture of the COVID-19 vaccines.

The findings of Table 6 on the subject area of published studies on the COVID-19 vaccines reveal that more than 19% of this research are in the *Immunology* area, which could probably indicate that researchers in this field, in addition to treating and producing vaccines for this epidemic, also examined the types of immune responses to these vaccines. Furthermore, *General & Internal Medicine* with 1054 and *Research Experimental Medicine* with 834 studies were other active areas in this field. In total, researchers in these three areas played an influential role in the research outputs on the COVID-19 vaccines by publishing more than 49 percent of the studies. Additionally, the situation of other active and productive areas in this field is presented in Table 6. In other words, studies on the COVID-19 vaccines have not been ineffective in broadening the horizons of specialists, researchers, and physicians to control the disease.

Figure 10 illustrates the research focus of researchers in the field of COVID-19 vaccines based on the cloud label. Cloud tags show the extent to which researchers pay attention to topics; the higher the frequency of keywords used in research by researchers, the larger the tags, and the lower the frequency, the smaller the tags [22]. Figure 10 illustrates topics, such as COVID-19, COVID-19 vaccination, COVID-19 vaccines, vaccine hesitancy, COVID-19 pandemic, public health, COVID-19 vaccines bnt162b2, COVID-19 vaccine safety, healthcare workers, and vaccine acceptance.

Figure 11 shows the trend of research on the COVID-19 vaccines based on the time zone view. In the linkage of Figure 11, the trend between nodes is moving from left to right, and in a way that indicates what topics the researchers have focused on and what topics they have tended to recently [23]. In addition, the time zone map represents the amount of attention of researchers to the topics in different time periods [23]. In 2019, topics, such as Spike protein, COVID-19, receptor binding domain, immunogenicity, safety of COVID-19 vaccines, infection, antibody, and neutralizing antibody are more visible in the researches of researchers in this period. The trend of research in 2020 confirms that in this period, researchers have reduced their research on the identification of this virus and focused their research on topics, such as spike, influenza COVID-19 vaccination, health, protection, COVID-19 vaccination, acute respiratory syndrome, respiratory syndrome COVID-19, immunization, influenza, immunity COVID-19 vaccines safety, disease, and behavior. However, the research trends of researchers in 2021 indicate that topics, such as COVID-19 infection, vaccine hesitancy COVID-19 vaccines, attitude, adult, and mortality have been more prominent than other topics in the studies of this period.

Figure 12 and Figure 13 show a linking map of countries and institutions. Based on this map, it is possible to examine the relationship between each country and institution, as well as the research fields that countries are studying. Figure 12 and Figure 13 not only show the communication status and research areas of countries and institutions, but also indicates the timeline of their studies in the field of the COVID-19 vaccine. Therefore, the time period of the study of countries and institutions in the field of the COVID-19 vaccine on the horizontal axis and the research fields of their studies on the vertical axis are shown. Based on Figure 12, it can be argued that the US, as one of the leading countries in the COVID-19 vaccine, started its activity in the field of vaccine production in 2019, and its activities have continued with more intensity in recent years, which is seen from the red and yellow colors around the ring of the US. Additionally, the fact that this country is in the #1 cluster can confirm that the US has not only paid attention to the issue of vaccine production but has also pursued the issue of sociodemographic factor.

England (the UK) and Germany, on the other hand, are other productive countries in this field, which is in the #1 cluster called the sociodemographic factor. China is also the source of the corona outbreak in cluster #2, and Italy and Russia in clusters #4 and #0 called the effective COVID-19 genomics and COVID-19 vaccines, are paying close attention to aspects of new strains and variants COVID-19 and COVID-19 vaccines.

Figure 13 also consists of twelve clusters. In Figure 13, the University of Oxford, Harvard Medical School, London School of Hygiene & Tropical Medicine, Tel Aviv University, and Washington University formed the core of research institutes related to the COVID-19 vaccines in the clusters of public health, COVID-19 vaccines, COVID-19 vaccines hesitancy, and COVID-19 vaccination, which in itself can indicate that these institutions have conducted extensive work in these areas. However, in the link map, the Chinese Academy of Sciences, and the University of Texas Medical Branch had the highest citation flourishing scores, and in a way, the research conducted by researchers from these institutes have been able to attract the most attention of the scientific community.

Table 7 shows that vaccines of AZD1222 (Oxford-AstraZeneca), BNT162b2 (Pfizer-BioNTech), Ad26.COV2.S (Johnson and Johnson), and mRNA-1273 (Moderna) have been injected by more countries. In total, these vaccines were licensed for use in 134, 132, 101, and 85 countries, respectively (See Appendix B). Additionally, the three BNT162b2, mRNA-1273, and AZD1222 vaccines with 940, 545, and 403 certificates, respectively, had the highest number of publications in the WoS database. It should be noted that according to the results of Table 7, China with 5, Russia and Iran each with 4, US with 3, and India with 2 vaccines, were the leading countries in vaccine production. Moreover, most of the values prepared are related to the two vaccines of BNT162b2 with 4,921,729,970 doses and mRNA-1273 with 3,078,314,353 doses.

## 4. Discussion

Due to the widespread prevalence of COVID-19 worldwide and its new mutations, many studies have been conducted by researchers on therapeutic and pharmacological methods to treat COVID-19 and to develop a vaccine to control the disease. According to the latest statistics approved by the World Health Organization, the number of patients with “COVID-19” disease in the world has reached 270,791,973 people, and the deaths of 5,318,216 people have been confirmed due to this disease. On 15 December 2021, more than 8,200,642,671 people worldwide had been vaccinated against the virus [9]. Therefore, the results indicate that by the end of 13 December 2021, 6288 studies related to the COVID-19 vaccines have been published in the Web of Science citation index.

The findings related to the trend of publications related to the COVID-19 vaccines indicate a significant increase compared to 2019 and 2020, meaning that more than 90 percent of these publications have been produced in 2021. Therefore, a reason for the significant increase in releasing research related to COVID-19 vaccines in 2021 may be the decline in global resilience faced with this epidemic and a more deadly strain of the virus, which prompted researchers to conduct further studies on the field of making and producing vaccines and finally controlling and eradicating this disease. Overall, the findings of researchers and institutions with the highest number of research publications showed that Pollard AJ and Lambe T of the University of Oxford had the highest number of publications on studies related to the COVID-19 vaccines. The findings from the most active research institutes also indicated that the University of Oxford in the UK and the Harvard Medical School in the US, as the most active institutions, played a more prominent role than other institutes in studies on the COVID-19 vaccines. The above results agree with the findings of Ay et al. [18].

The findings from leading countries in studies on the COVID-19 vaccines showed that researchers from the US published the most publications in this field; the results of Surulinathi et al. [24] and Ay et al. [18] confirm the findings of this study. The UK, Italy, and China also played a key role in the dissemination of the scientific documents. In other words, researchers from these four countries together played an important role in this field by publishing more than 61 percent of the research. The results of scientific collaboration between countries also showed that the highest level of scientific collaboration has been formed between researchers from the US–UK and the US–China. Note that countries that had the most scientific collaborations were countries that could publish most publications in this field.

The results of the journals that published research on the COVID-19 vaccines showed that the researchers published their scientific publications in 1321 journals. Among them, the *Vaccines* and *Vaccine* journals published the largest number of studies on the COVID-19 vaccines during the desired period. These journals are published by MDPI and Elsevier publications, respectively, and in terms of ranking of the *Web of Science* journals, they are ranked Q2 and Q3, respectively, and are in a good position in terms of impact factor. However, a review of support organizations for COVID-19 vaccine studies showed that the U.S. Department of Health & Human Services, with 437 studies, and the National Institutes of Health (NIH) from the US, with 408 studies, provided the most support from published studies on the COVID-19 vaccines. Additionally, the results related to the thematic fields of published studies showed that researchers in the field of *Immunology* with 1244 studies (19.8%), the field of *General & Internal Medicine* with 1054 studies (16.8%), and the field of *Research Experimental Medicine* with 834 studies (13.3%) percentage) had the most scientific publications in this field, which in the study Ay et al. [18], these three subject areas were also the leading thematic fields in studies on COVID-19 vaccines.

The state of research on the COVID-19 vaccines based on the time zone map showed that the initial studies in 2019 were more focused on identifying the COVID-19 virus and ways to combat it (COVID-19 vaccines efficacy, immunogenicity). However, the scientific research of researchers in 2020 has led to topics, such as COVID-19 and vaccines (COVID-19 vaccination, COVID-19 vaccines safety, and influenza COVID-19 vaccination), which indicates their serious attention to the development of vaccines to control and treat the disease. Finally, the research trend of studies in 2021 on topics, such as COVID-19 vaccines and their types and effects, vaccines hesitancy, the role of healthcare workers in COVID-19, as well as discussions on these vaccines on social media has been more than other studies.

Findings on the interconnected map of countries and research institutes showed that the US has paid serious attention to scientific activities about the production of COVID-19 vaccine, vaccination, and sociodemographic factor. The universities of Oxford, Harvard Medical School, Tel Aviv, Washington, and the London School of Hygiene & Tropical Medicine have also been major research institutes in the field of COVID-19 vaccines.

## 5. Conclusions

Generally, this study examined the trend of publications on the COVID-19 vaccines from the perspective of scientometric and depicted the topics and fields of study of countries, institutions, and researchers and the collaboration between them in this field. The findings of this study can be valuable through the identification and analysis of the most important actors (country, institution, researcher, and channel of release of COVID-19 vaccines), the research process and fields of study in the COVID-19 vaccine for researchers, countries, and policymakers in the fields of medical sciences and health to make better decisions and understand and ultimately progress in this field. There may also be more research in the future on other types of vaccines and comparing their effectiveness on mutated strains of the virus, vaccine acceptance, long-term side effects of these vaccines, COVID-19 vaccination from various perspectives, and other related topics due to the more prominent keywords, such as “COVID-19 vaccine” and “COVID-19 vaccination”, which can also increase the upward trend of publication compared to recent years. Finally, there is a need to increase opportunities for financial support, more scientific collaborations, understanding and analyzing research of leading countries, and emulating them from different angles to make appropriate decisions.

## Figures and Tables

**Figure 1 jcm-11-00750-f001:**
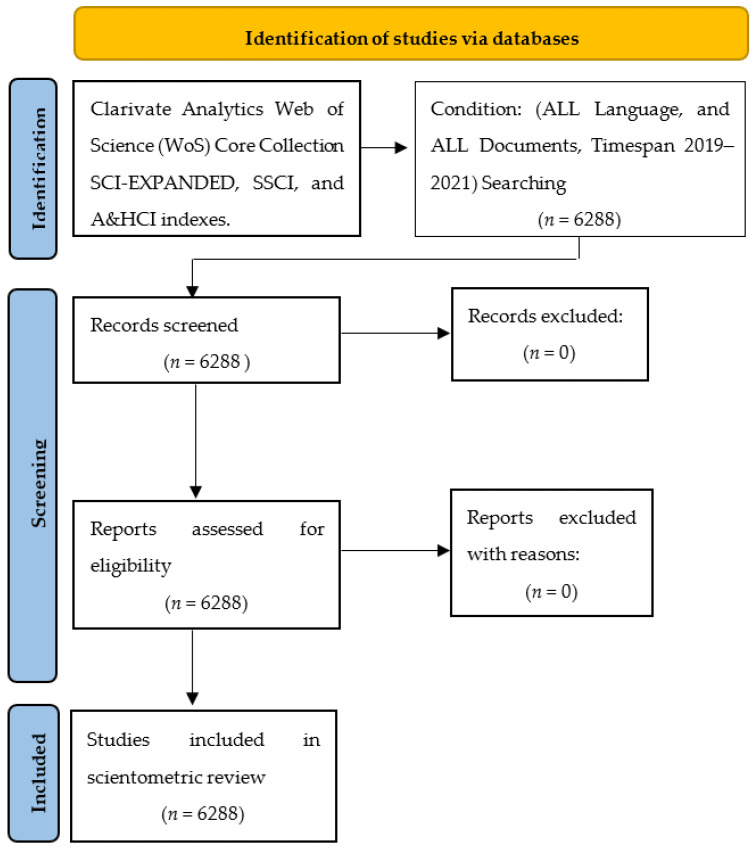
PRISMA flow diagram in COVID-19 vaccines.

**Figure 2 jcm-11-00750-f002:**
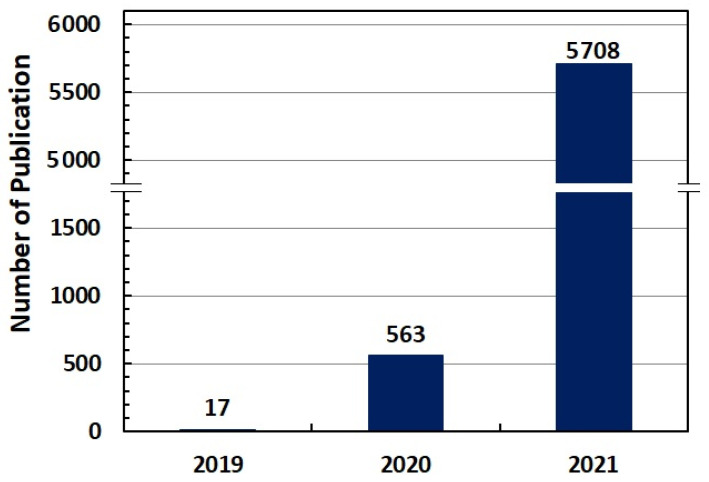
The Annual distribution of in COVID-19 vaccine research.

**Figure 3 jcm-11-00750-f003:**
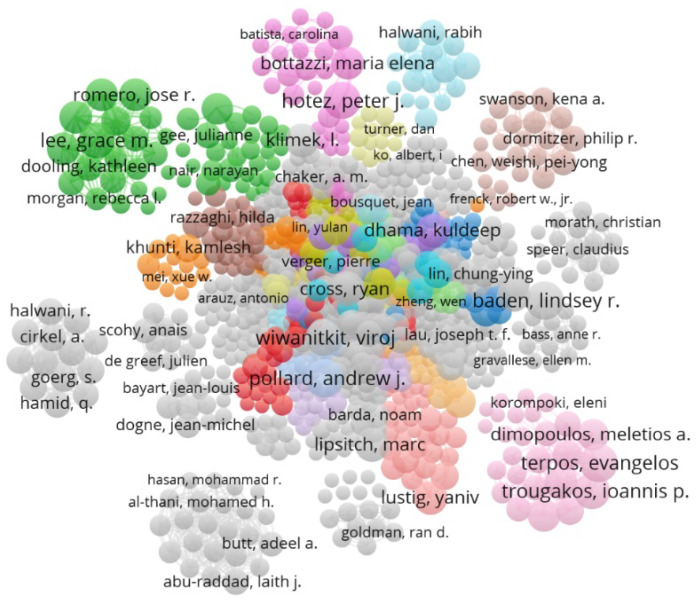
Authors’ collaboration network in COVID-19 vaccine research.

**Figure 4 jcm-11-00750-f004:**
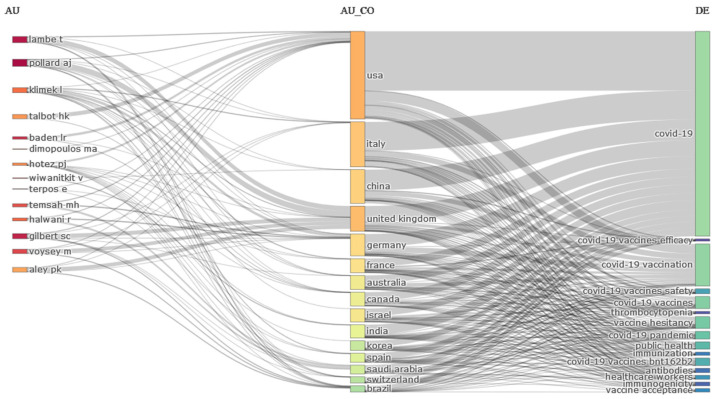
Three-field plot for the relationships among authors, countries, and author keywords in COVID-19 vaccine research. (AU: Author, Au_CO: Author country, DE: Document keyword).

**Figure 5 jcm-11-00750-f005:**
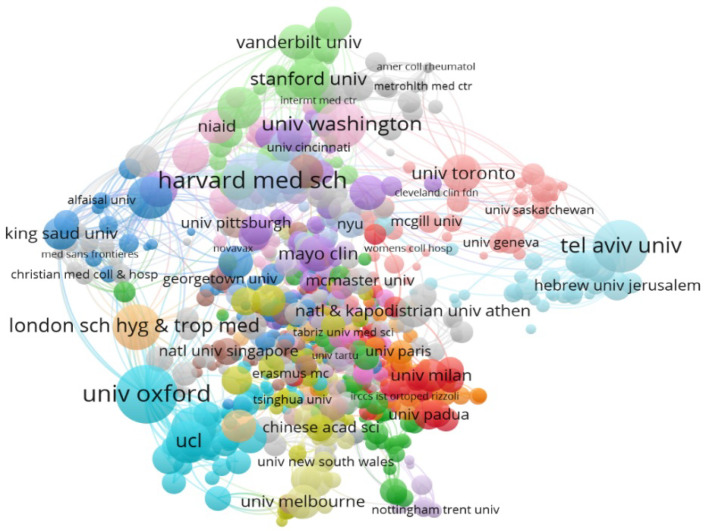
Institutional collaboration network in COVID-19 vaccine research.

**Figure 6 jcm-11-00750-f006:**
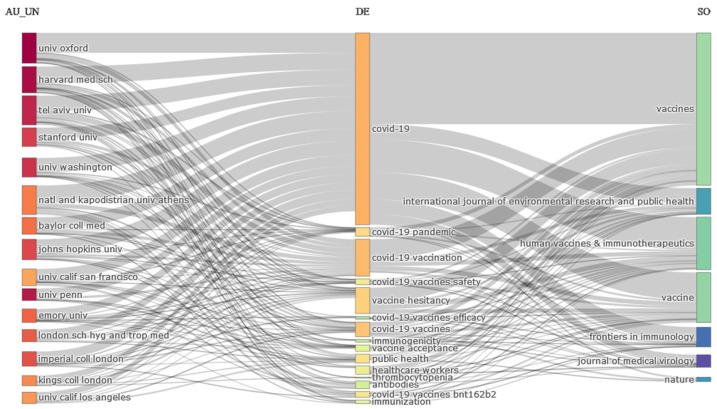
Three-field plot of the relationships between institutions, authors’ keywords, and sources in the research of the COVID-19 vaccine. (AU_UN: Author university, DE: keyword, SO: Source).

**Figure 7 jcm-11-00750-f007:**
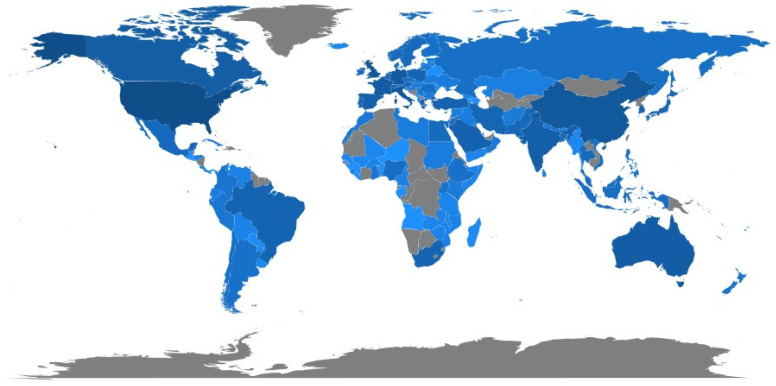
The Geographical map of countries prolific in COVID-19 vaccine research (Dark blue color: countries with a high number of articles related to the COVID-19 vaccine, light blue color: countries with a low number of articles related to the COVID-19 vaccine, gray color: countries without articles related to the COVID-19 vaccine).

**Figure 8 jcm-11-00750-f008:**
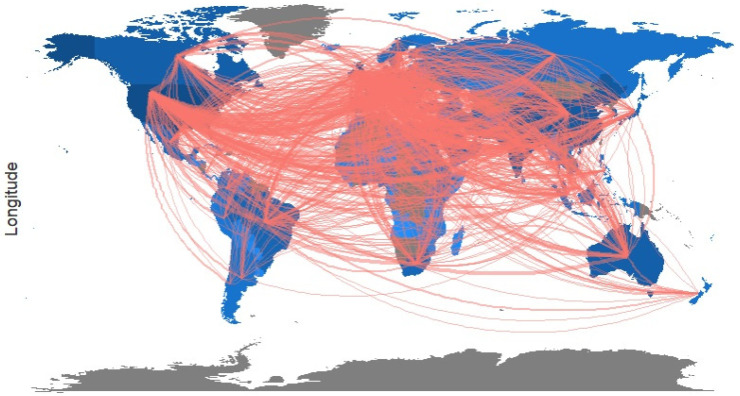
The Geographical map of the collaboration of countries in the research of the COVID-19 vaccine research (Dark blue color: countries with a high number of articles related to the COVID-19 vaccine, light blue color: countries with a low number of articles related to the COVID-19 vaccine, gray color: countries without articles related to the COVID-19 vaccine).

**Figure 9 jcm-11-00750-f009:**
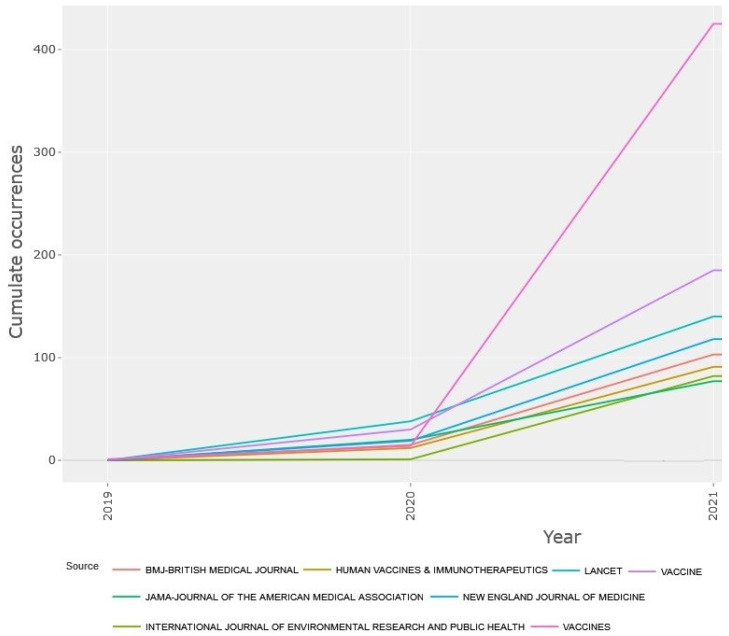
Annual occurrences of the most prolific journals in COVID-19 vaccine research.

**Figure 10 jcm-11-00750-f010:**
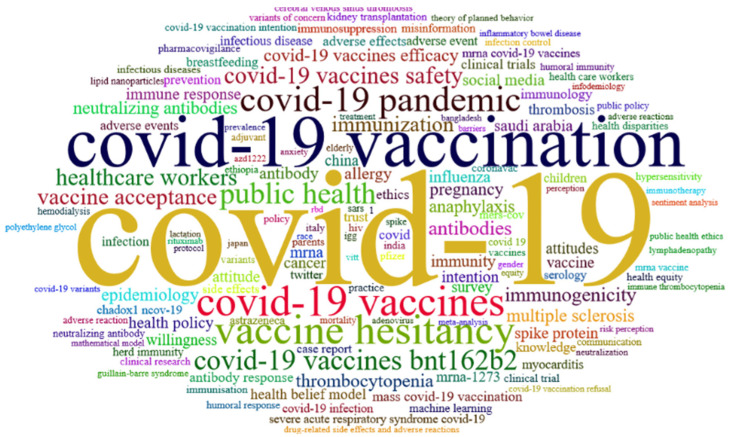
Word cloud map authors keywords in COVID-19 Vaccine research.

**Figure 11 jcm-11-00750-f011:**
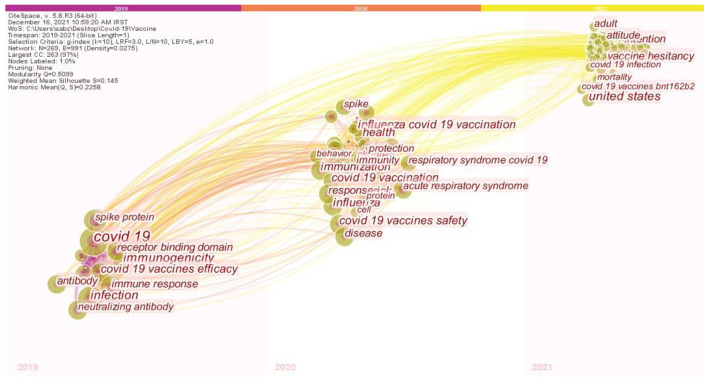
The time zone visualization map of co-occurring keywords in COVID-19 Vaccine research.

**Figure 12 jcm-11-00750-f012:**
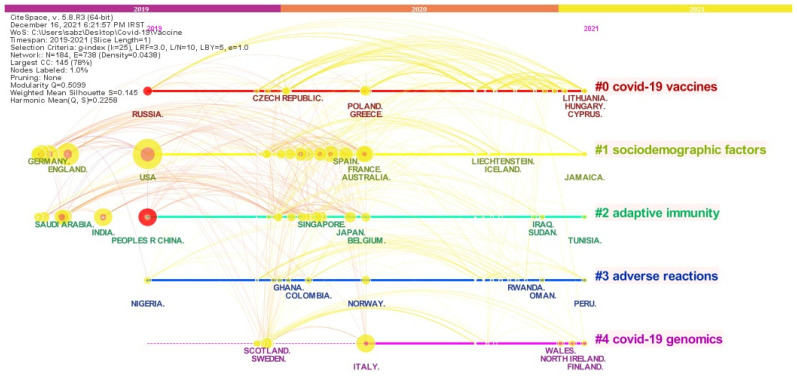
Timeline map of countries in COVID-19 vaccines research based on Authors keywords.

**Figure 13 jcm-11-00750-f013:**
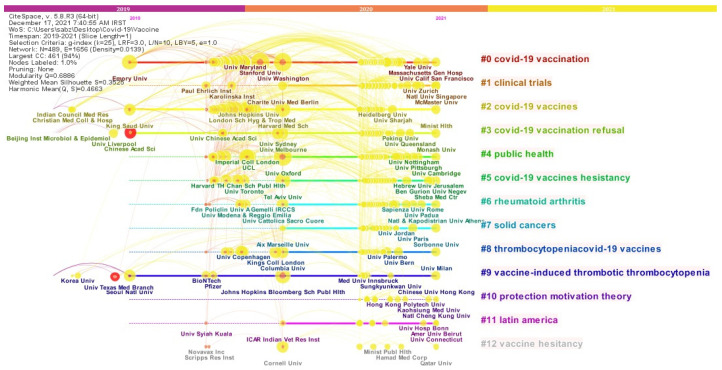
Timeline map of institutions in COVID-19 vaccines research based on Authors keywords.

**Table 1 jcm-11-00750-t001:** The most prolific authors in COVID-19 vaccine research. (NP: Number of Publications, TC: Total Citations).

Rank	Authors	H-Index	G-Index	HC-Index	NP	TC	PY_Start
1	Pollard AJ	13	36	17	36	3583	2020
2	Lambe T	13	29	18	29	3579	2020
3	Gilbert SC	11	24	16	24	3238	2020
4	Baden LR	4	23	6	23	1823	2020
5	Wiwanitkit V	1	4	1	23	21	2021
6	Voysey M	10	22	13	22	3210	2020
7	Temsah MH	3	4	4	20	19	2021
8	Terpos E	6	10	11	20	118	2021
9	Halwani R	3	3	4	19	14	2021
10	Klimek L	4	8	8	19	81	2021
11	Dimopoulos MA	6	10	10	18	115	2021
12	Hotez PJ	8	18	12	18	400	2020
13	Talbot HK	10	18	13	18	599	2020

**Table 2 jcm-11-00750-t002:** The most prolific institutions in COVID-19 vaccine research.

Rank	Institutions	NP	TC	Country	Location
1	University of Oxford	151	5546	UK	Oxford
2	Harvard Medical School	144	2050	US	Boston
3	Tel Aviv University	119	1633	Israel	Tel Aviv
4	University of Washington	99	2128	US	Seattle, Washington
5	London School of Hygiene & Tropical Medicine	87	3171	UK	Bloomsbury, London
6	University College London	83	2993	UK	London
7	Imperial College London	80	2331	UK	London
8	Johns Hopkins University	79	1328	US	Baltimore, Maryland
9	University of Pennsylvania	78	754	US	Philadelphia, Pennsylvania
10	Emory University	76	4008	US	Atlanta, Georgia
11	Stanford University	70	963	US	Stanford, California
12	University of Toronto	60	422	Canada	Toronto

**Table 3 jcm-11-00750-t003:** The most prolific countries in COVID-19 vaccine research.

Rank	Countries	NP	TC
1	US	2104	32,958
2	UK	776	15,269
3	Italy	521	3078
4	China	435	5671
5	Germany	391	9389
6	India	280	1836
7	Canada	263	2869
8	Australia	259	2003
9	France	247	1871
10	Israel	217	3000
11	Spain	205	1589
12	Switzerland	172	1702

**Table 4 jcm-11-00750-t004:** The most prolific journals in COVID-19 vaccine research. (IF: Impact factor, Q: Quarterly journal).

Rank	Journal	NP	TC	IF	Q	Publisher
1	Vaccines	425	2450	4.422	2	MDPI
2	Vaccine	185	1745	3.641	3	Elsevier
3	Human Vaccines & Immunotherapeutics	174	664	3.452	3	Taylor & Francis
4	Lancet	140	6776	79.321	1	Elsevier
5	New England Journal of Medicine	135	12861	91.245	1	Massachusetts Medical Society
6	British Medical Journal	103	679	17.215	1	BMJ
7	Jama-Journal of the American Medical Association	85	1744	56.272	1	American Medical Association
8	International Journal of Environmental Research and Public Health	82	253	3.39	2	MDPI
9	Nature	61	2621	49.962	1	Nature Publishing Group
10	Journal of the European Academy of Dermatology and Venereology	59	231	6.166	1	Wiley
11	MMWR-Morbidity and Mortality Weekly Report	59	1414	17.586	1	Centers for Disease Control and Prevention
12	Frontiers in Immunology	54	446	7.561	1	Frontiers Media S.A.
13	Lancet Infectious Diseases	53	1239	25.071	1	Lancet Publishing Group
14	Nature Medicine	51	1807	53.44	1	Nature Publishing Group

**Table 5 jcm-11-00750-t005:** Active funding agencies in COVID-19 vaccine research.

Rank	Funding Agencies	Record Count	% of 6288
1	US Department of Health & Human Services	437	6.9%
2	National Institutes of Health (NIH) US	408	6.5%
3	European Commission	169	2.7%
4	UK Research Innovation UKRI	143	2.3%
5	NIH National Institute of Allergy Infectious Diseases NIAID	119	1.9%
6	National Natural Science Foundation of China NSFC	114	1.8%
7	Medical Research Council UK MRC	92	1.4%
8	National Institute For Health Research NIHR	78	1.2%
9	Bill Melinda Gates Foundation	67	1.1%
10	National Science Foundation NSF	63	1.0%

**Table 6 jcm-11-00750-t006:** Active research areas in COVID-19 vaccine research.

Rank	Research Areas	Record Count	% of 6288
1	Immunology	1244	19.8%
2	General & Internal Medicine	1054	16.8%
3	Research Experimental Medicine	834	13.3%
4	Public Environmental Occupational Health	627	10.0%
5	Infectious Diseases	382	6.1%
6	Science Technology Other Topics	298	4.7%
7	Pharmacology Pharmacy	295	4.7%
8	Multidisciplinary Sciences	259	4.1%
9	Neurosciences & Neurology	247	3.9%
10	Biotechnology Applied Microbiology	215	3.4%

**Table 7 jcm-11-00750-t007:** Information on COVID-19 vaccines, type of vaccine, procured dose, approval satus, publications in WoS.

Name of Vaccine	Primary Developers ^1^	Type of Vaccine ^1^	Country of Origin ^2^	No. Procured Doses ^3^	Status of Approving Countries ^1^	No. Publications WoS
AZD1222	Oxford-AstraZeneca	Non-Replicating Viral Vector	UK	2,039,354,959	134	403
BNT162b2	Pfizer-BioNTech	RNA	Multinational	4,921,729,970	132	940
mRNA-1273	Moderna	RNA	US	3,078,314,353	85	545
Ad26.COV2.S	Johnson & Johnson	Non-Replicating Viral Vector	US/Netherlands	1,327,811,478	101	18
BBIBP-CorV	Sinopharm (Beijing)	Inactivated	China	787,536,122	85	44
WIBP-CorV	Sinopharm (Wuhan)	Inactivated	China	4,420,000	2	1
Sputnik V	Gamaleya	Non-Replicating Viral Vector	Russia	243,543,406	74	44
Sputnik Light	Gamaleya	Non-Replicating Viral Vector	Russia	889,800	24	-
CoviVac	Chumakov Center	Inactivated	Russia	1,356,000	3	2
CoronaVac	Sinovac	Inactivated	China	1,150,377,05	51	86
Covaxin (BBV152)	Bharat Biotech	Inactivated	India	384,620,00	13	26
Soberana 02	Finlay de Vacunas	Protein Subunit	Iran/Cuba	16,000,000	4	2
NVX-CoV2373	Novavax	Protein Subunit	US	870,418,000	31	30
COVAX-19 (Spikogen)	CinnaGen	Protein Subunit	Iran	12,000,000	1	1
Ad5-nCoV (Convidicea)	CanSino	Non-Replicating Viral Vector	China	23,853,699	10	13
COVIran Barekat	Shifa Pharmed	Inactivated	Iran	300,000,000	1	1
EpiVacCorona	FBRI	Protein Subunit	Russia	1800	4	2
FakhraVac (MIVAC)	Organization of Defensive Innovation and Research	Inactivated	Iran	Not Available	1	1
QazVac	Kazakhstan RIBSP	Inactivated	Kazakhstan	Not Available	2	2
ZyCoV-D	Zydus Cadila	DNA	India	Not Available	1	3
MVC-COV1901	Medigen	Protein Subunit	Tiwan	Not Available	2	2
ZF2001	Anhui Zhifei Longcom	Protein Subunit	China/Uzbekistan	Not Available	3	4

References: ^1^ primary developers, Type of vaccine and Status of approving countries (https://covid19.trackvaccines.org/vaccines/approved/) ^2^ Country of Origin (https://www.raps.org/news-and-articles/news-articles/2020/3/covid-19-vaccine-tracker) ^3^ No. procured doses (https://app.powerbi.com/view?r=eyJrIjoiMWNjNzZkNjctZTNiNy00YmMzLTkxZjQtNmJiZDM2MTYxNzEwIiwidCI6ImY2MTBjMGI3LWJkMjQtNGIzOS04MTBiLTNkYzI4MGFmYjU5MCIsImMiOjh9). (All data extracted in 19 January 2022).

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
