# Peer review of "Current and Future Perspectives on the COVID-19 Vaccine: A Scientometric Review"

_jcm, 2022, doi:10.3390/jcm11030750_

Round 1

Reviewer 1 Report

The manuscript “Current and Future Perspectives on the COVID-19 Vaccine: A 2 Scientometric Review”  attempted to draw the present and future perspective of the COVID-19 vaccine 21 by identifying the most important scientists and their scientific contexts, trends of research topics, 22 and relationships between different entities.

Although the general idea of the manuscript is of potential interest, the article is the manuscript is really an original scientometry and meta-analysis article rather than a review.

The methodology presented is not convincing and is incomplete, should be expanded, detailing all the sources used, providing more information on the methods.  

Information on vaccines in preclinical development should be expanded.

Annex A is confusing and should be reassembled.

Reviewer 2 Report

In this review, the authors collected data related to Scientometric indexes for COVID-19 vaccine.  The review includes the followings: the annual distribution of in COVID-19 vaccine research during 2019-2021, authors' collaboration network in COVID-19 vaccine research, institutional collaboration network,most prolific countries in the vaccine research, and most prolific journals.

I would suggest more points to be included

a) Name/ type of  COVID19 vaccines used, doses.

b) If side effects mentioned in the mentioned papers

c) link between the source of COVID19 vaccines, country of manufacture, and the research papers included.

d)link between the source of COVID19 vaccines, country of manufacture, and the country of publisher.

Round 2

Reviewer 1 Report

The authors have successfully modified the manuscript. 

Reviewer 2 Report

The authors addressed my questions